# Casein kinase 1 gamma regulates oxidative stress response via interacting with the NADPH dual oxidase complex

Yiman Hu[1], Zhaofa Xu[1], Qian Pan[1], Long Ma[1,2,3,4]*

1 Center for Medical Genetics, School of Life Sciences, Central South University, Changsha, Hunan, China, 2 Hunan Key Laboratory of Animal Models for Human Diseases, Central South University, Changsha, Hunan, China, 3 Hunan Key Laboratory of Medical Genetics, Central South University, Changsha, Hunan, China, 4 The Key Laboratory of Precision Molecular Medicine of Hunan Province, Central South University, Changsha, Hunan, China

* malong@sklmg.edu.cn

**Data Availability Statement:** All relevant data are within the manuscript and its Supporting information files.

**Funding:** The study is supported by National Natural Science Foundation of China grants (No.

## Abstract

Oxidative stress response is a fundamental biological process mediated by conserved mechanisms. The identities and functions of some key regulators remain unknown. Here, we report a novel role of *C. elegans* casein kinase 1 gamma CSNK-1 (also known as CK1γ or CSNK1G) in regulating oxidative stress response and ROS levels. *csnk-1* interacted with the *bli-3/tsp-15/doxa-1* NADPH dual oxidase genes via genetic nonallelic noncomplementation to affect *C. elegans* survival in oxidative stress. The genetic interaction was supported by specific biochemical interactions between DOXA-1 and CSNK-1 and potentially between their human orthologs DUOXA2 and CSNK1G2. Consistently, CSNK-1 was required for normal ROS levels in *C. elegans*. CSNK1G2 and DUOXA2 each can promote ROS levels in human cells, effects that were suppressed by a small molecule casein kinase 1 inhibitor. We also detected genetic interactions between *csnk-1* and *skn-1* Nrf2 in oxidative stress response. Together, we propose that CSNK-1 CSNK1G defines a novel conserved regulatory mechanism for ROS homeostasis.

## Author summary

Oxidative stress response is a fundamental life process that regulates the generation and elimination of reactive oxygen species (ROS) and manages ROS-elicited damages. Some key regulatory mechanisms remain to be elucidated. While analyzing mutations promoting the survival of the nematode *Caenorhabditis elegans* in excess iodide, an oxidative stressor, we identified the casein kinase 1 gamma gene *csnk-1*. Casein kinase 1 gamma proteins are highly conserved and proposed to have important biomedical functions. However, they are understudied and their effects on oxidative stress response are unknown. We combined genetic, biochemical and cell biology approaches to show that casein kinase 1 gamma members in *C. elegans* and humans are novel regulators of oxidative stress response and promoters of ROS levels. These activities involve interactions with the NADPH dual oxidase complex, which are likely mediated by physical interactions

31972877) and Natural Science Foundation of Hunan Province (2020JJ4109) to LM. The funders had no role in study design, data collection and analysis, decision to publish, or preparation of the manuscript.

**Competing interests:** The authors have declared that no competing interests exist.

between casein kinase 1 gamma and the dual oxidase maturation factor. Our findings provide new mechanistic insights into the regulation of ROS homeostasis and uncover an important function of the "dark" casein kinase 1 gamma.

## Introduction

Reactive oxygen species (ROS) are universal life molecules. ROS at physiological levels are essential signaling molecules that regulate metabolism, cell growth, cell survival and cell proliferation [1]. Endogenous ROS are primarily generated by the NADPH oxidases and mitochondria [1,2]. Abnormal cellular processes, xenobiotics, some metals and pathogens can induce excessive ROS generation [3–5], which may damage proteins, lipids and nucleic acids and cause oxidative stress [2]. Animals use conserved molecular mechanisms to regulate ROS levels and mitigate oxidative stress [1,2]. For example, the nuclear factor erythroid 2-related factor 2 (Nrf2) pathway plays a major role in the defense response to oxidative stress triggered by endogenous or exogenous stressors [1,2,6].

Studies in the nematode *Caenorhabditis elegans* have provided important mechanistic understandings of oxidative stress response [6,7]. *C. elegans* genome encodes a Nrf2 homolog SKN-1 [8] that mediates oxidative stress response [9] and is also broadly involved in aging, immunity, lipid metabolism and other stress responses [6]. SKN-1 activity is regulated by p38 MAPK, GSK-3 and other signals [6,10,11]. Similar to the mammalian Keap1-mediated ubiquitination and proteasome degradation of Nrf2 [12], SKN-1 levels can be negatively regulated by the WD40 repeat protein WDR-23 via the ubiquitin-proteosome pathway [13].

*C. elegans* dual oxidase BLI-3 is the only functional NADPH oxidase encoded by its genome [14]. BLI-3, the tetraspanin protein TSP-15 and the dual oxidase maturation factor DOXA-1 form a NADPH oxidase complex to regulate *C. elegans* cuticle formation and response to oxidative stress [7,14–20]. BLI-3 affects innate immunity by interacting with p38 MAPK, SKN-1, proline catabolism and DAF-16 pathways [21–27]. BLI-3 also plays roles in longevity [15,28], vulva development [29] and the response to manganese toxicity [30]. The mammalian orthologs of BLI-3, DUOX1 and DUOX2, are involved in immunity and required for thyroid hormone synthesis, respectively [31].

Casein kinase 1 (CSNK1) is a family of conserved serine/threonine kinases that regulates diverse signaling processes by phosphorylating a large number of protein substrates [32]. Mammals have six common casein kinase 1 members, CSNK1A (CK1α), CSNK1D (CK1δ), CSNK1E (CK1ε) and CSNK1G1/2/3 (CK1γ1/2/3) [33], among which CSNK1Gs require palmitoylation-mediated membrane localization for their activities [34–36]. Studies in *Drosophila* and mammals found that CSNK1Gs can regulate Wnt, Hedgehog, JNK and RIPK3 signals [34,36–39], and can also phosphorylate the ceramide transport protein [40] and Lyn tyrosine kinase [41]. Human CSNK1G1 mutations are associated with developmental delay and autism spectrum disorder [42].

In *C. elegans*, *kin-19*, *kin-20* and *csnk-1* encode orthologs of CSNK1A, CSNK1D and CSNK1G, respectively [43] (wormbase.org). *csnk-1* is an essential gene for *C. elegans* embryonic asymmetric spindle positioning [44–48] and oocyte meiosis [49]. The function of CSNK1G in oxidative stress response is unclear.

Iodine is an essential mineral nutrient for the synthesis of thyroid hormones and is routinely added to salt and foods. Surprisingly, some studies found that excess intake of iodide salt was associated with several thyroid diseases with unclear molecular mechanisms [50]. We became interested in this phenomenon and used *C. elegans* as a model to examine the

biological effects of excess iodide intake. We found that *C. elegans* treated with excess iodide had a significant increase of ROS levels and wildtype animals exhibited larval arrest potentially due to oxidative stress [19,20]. This is likely a conserved effect of iodide because several earlier studies showed that excess iodide can induce ROS generation in mammalian thyroid cells [51–54]. To investigate whether any genes might be involved in the larva-arresting effect of excess iodide, we screened for mutants that can survive into adults in excess iodide and isolated multiple loss-of-function (lf) mutations in *bli-3*, *tsp-15*, *doxa-1*, *wdr-23* and gain-of-function (gf) mutations in *skn-1* [19,20]. These findings suggest that new genes involved in oxidative stress response might be identified by examining whether their mutations can promote *C. elegans* survival in excess iodide [19,20].

In this study, we found that *mac397*, a previously isolated mutation that promoted *C. elegans* survival in excess iodide [20], caused a missense mutation in *csnk-1*. Further analyses found that *csnk-1* genetically interacted with the *bli-3/tsp-15/doxa-1* genes by nonallelic non-complementation to affect animal survival in excess iodide. Consistent with this special genetic interaction, we detected specific biochemical interactions between CSNK-1 and DOXA-1 and potentially between their human orthologs. We also found that CSNK-1 was required for maintaining ROS levels in *C. elegans* and human CSNK1G2 can promote ROS levels in cultured cells. We propose that CSNK-1 CSNK1G is a conserved novel regulator of ROS levels and oxidative stress response by interacting with the NADPH dual oxidase complex.

## Results

### *csnk-1* loss of function promotes *C. elegans* survival in excess iodide

We previously performed an ethyl methanesulfonate (EMS) screen for F$_1$ mutants that can grow into adults in 5 mM NaI (the survived-in-sodium-iodide phenotype, abbreviated as Sisi hereafter) and isolated six unidentified mutants [20]. We mapped one of the mutations, *mac397*, to Chr. I. Within the mapped region of *mac397*, genomic sequencing detected a point mutation in *csnk-1* (exon4:c.G482A:p.R161H, named *mac397*. Fig 1A and S1 Table), a point mutation in *tsp-15* (exon3:c.G331A:p.G111R, named *mac499*. S1 Table), and nine other coding variants (S1 Table). Deficiencies in *csnk-1* or *tsp-15* caused by RNAi or genetic mutations led to the Sisi phenotype (S1, S2 and S3 Tables). Transgene rescue and genocopy experiments suggest that *tsp-15(mac499)* caused a loss of function (S3 Table). The analyses of other isolates are ongoing.

*csnk-1* encodes a casein kinase 1 gamma [43] orthologous to the mammalian CSNK1G1, CSNK1G2 and CSNK1G3 [33]. The kinase domains and the C-terminal palmitoylation signals of CSNK1Gs are highly conserved (S1 Fig). The R161H (arginine to histidine) change caused by *mac397* is in the kinase domain of CSNK-1 (S1 Fig).

To verify the effect of *csnk-1* on the Sisi phenotype, we generated two frameshift mutations, *mac494* and *mac495* (Fig 1A and S2 Table), using the CRISPR/Cas9 method. Homozygous progeny of *csnk-1(mac494)* or *csnk-1(mac495)* heterozygotes exhibited the Sisi phenotype, while the heterozygous progeny did not (Fig 1B and S2 Table).

To investigate how *csnk-1* transcript levels change in these mutants, we performed RT-qPCR experiments using primer pairs that are supposed to recognize all transcripts (wildtype and mutants) or wildtype-only transcripts (S2A Fig). We found that both mutants exhibited significantly reduced levels of total *csnk-1* transcripts at the L4 larval stage (S2B Fig), implying a positive feedback effect of CSNK-1 on the expression of its own transcripts or that the mutant transcripts were degraded via nonsense-mediated decay (NMD). Furthermore, wildtype transcripts were essentially depleted in both mutants at the L4 larval stage (S2C and S2D Fig).

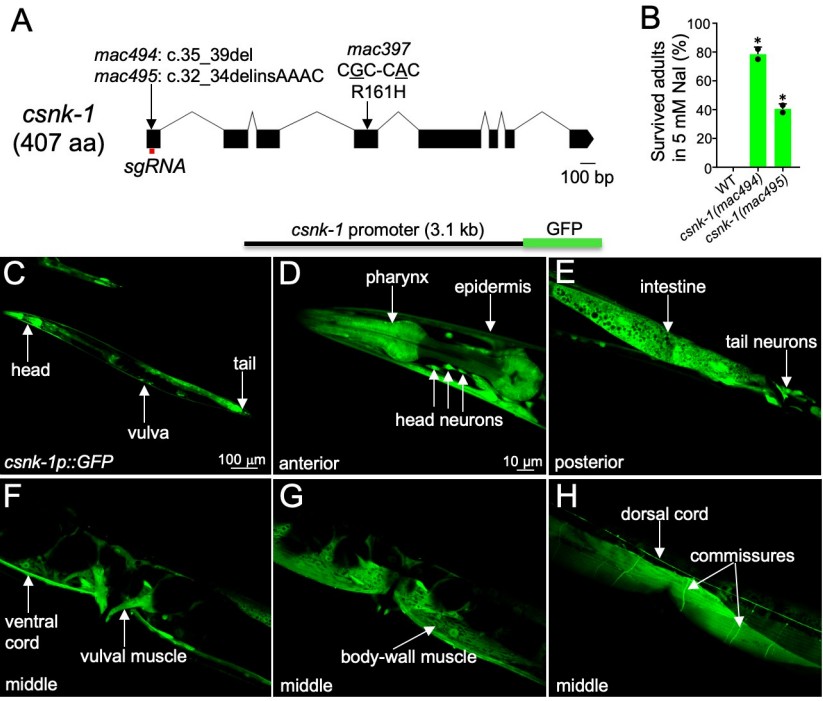

**Fig 1. Characterization of *csnk-1*.** (A) *csnk-1* gene structure (based on wormbase.org). The positions of *mac397*, *mac494* and *mac495* mutations are indicated. The position of the *sgRNA* used for CRISPR/Cas9-based mutagenesis is shown as a red bar. (B) Percentage of L1 larva that grew into adults on plates with 5 mM NaI. Results were based on two biological replicates. 100 L1 larva were analyzed in each replicate. Statistics: two-tailed unpaired Student's *t*-test. *: $p < 0.05$. (C-H) Expression pattern of a *csnk-1p::GFP* transgene in adults. (C) Fluorescent picture of a transgenic adult. The head, tail and vulva are indicated. (D-H) Higher-resolution pictures of transgenic adults. Cells with obvious GFP expression are indicated.

We observed other phenotypes of these mutants. For example, homozygotes can lay plenty of eggs (S3A Fig), but few would hatch (S3B Fig). *csnk-1* mutants exhibited significant molting defects (S3C–S3E Fig). However, if the molting was normal, the mutants exhibited grossly regular cuticle stripe patterns (S3F and S3G Fig). Like wild type, *csnk-1(mac494)* mutants were rarely stained by the nuclear dye Hoechst 33258 (S3H Fig), implying relatively normal cuticle integrity.

The egg-hatching, egg-laying and molting defects of *csnk-1(mac495)* mutants were obviously more severe than those of *csnk-1(mac494)* mutants. Since *mac494* and *mac495* both caused frameshifts close to the start codon and are predicted to encode severely truncated proteins (Figs 1 and S1), it is intriguing that the severity of their phenotypes was quite different. We have yet to determine whether this difference is caused by a potential hypomorphic nature of *mac494* or unknown modifier mutations in them.

To further understand *csnk-1* function, we expressed a *csnk-1* cDNA transgene driven by an endogenous *csnk-1* promoter (3.1 kb upstream of the start codon) in *csnk-1(mac494)* mutants. The transgene significantly rescued the Sisi phenotype of the mutants (Table 1). Because the majority of *csnk-1(mac494)* mutants were able to survive into adults probably due to apparently weaker molting defects (S3E Fig), which makes our analyses of the Sisi phenotype significantly easier, we used this mutation as our reference loss-of-function (lf) allele in the following studies, if not specified.

We next examined how *mac397* R161H affects CSNK-1 activity. Driven by the endogenous *csnk-1* promoter, a *csnk-1(R161H)* cDNA transgene failed to rescue or only weakly rescued the

**Table 1. Effects of *csnk-1* transgenes on the Sisi phenotype of *csnk-1(lf)* mutants in 5 mM NaI.**

| Background | Transgene | Tg line | Survival rate in 5 mM NaI (%) (n) | | | |
|---|---|---|---|---|---|---|
| | | | Experiment 1 | Experiment 2 | Average | *p*-value (vs *csnk-1(mac494lf)*) |
| WT | No *Tg* | - | 0% (100) | 0% (100) | 0.0% | - |
| | *csnk-1p::csnk-1_cDNA(p.R161H)* | 1 | 0% (100) | 0% (100) | 0.0% | - |
| | | 2 | 0% (100) | 0% (100) | 0.0% | |
| *csnk-1 (mac494lf)* | No *Tg* | - | 78%(100) | 80% (100) | 79.0% | - |
| | *csnk-1p::csnk-1_cDNA* | 1 | 6% (100) | 13% (100) | 9.5% | 0.002732 ** |
| | | 2 | 3% (100) | 5% (100) | 4.0% | 0.000355 *** |
| | *csnk-1p::csnk-1_cDNA(p.R161H)* | 1 | 81% (100) | 74% (100) | 77.5% | 0.720249 |
| | | 2 | 62% (100) | 56% (100) | 59.0% | 0.024100 * |
| | *dpy-7p::csnk-1_cDNA* (epidermis promoter) | 1 | 0% (100) | 1% (100) | 0.5% | 0.000203 *** |
| | | 2 | 5% (100) | 1% (100) | 3.0% | 0.000865 *** |
| | *nhx-2p::csnk-1_cDNA* (intestine promoter) | 1 | 77.6% (85) | 78.3% (83) | 78.0% | 0.426110 |
| | | 2 | 63.2% (76) | 72.7% (77) | 68.0% | 0.150572 |
| | *dpy-7p::HsCSNK1G1_cDNA* | 1 | 20% (100) | 18.8% (80) | 17.5% | 0.001911 ** |
| | | 2 | 16.8% (113) | 23% (100) | 19.9% | 0.003024 ** |
| | *dpy-7p::HsCSNK1G2_cDNA* | 1 | 10% (100) | 5% (100) | 7.5% | 0.001415 ** |
| | | 2 | 12% (100) | 6% (100) | 9.0% | 0.002035 ** |
| | *dpy-7p::HsCSNK1G3_cDNA* | 1 | 23% (100) | 21% (100) | 22.0% | 0.000615 *** |
| | | 2 | 42% (100) | 37% (100) | 39.5% | 0.004615 ** |
| | *csnk-1p::csnk-1_cDNA(p.C388_K407del)* | 1 | 60% (100) | 58% (100) | 59.0% | 0.004963 ** |
| | | 2 | 74% (100) | 63% (100) | 68.5% | 0.201122 |
| | *csnk-1p::csnk-1_cDNA(p.C388-390S)* | 1 | 56.2% (105) | 67% (100) | 63.0% | 0.060448 |
| | | 2 | 72% (100) | 78% (100) | 75.0% | 0.333333 |

The wildtype phenotype is non-survival in 5 mM NaI. n: total number of L1 larva examined. Statistics: two-tailed unpaired Student's *t*-test.

Sisi phenotype of *csnk-1(lf)* mutants (Table 1). This transgene also failed to cause Sisi in wild-type animals, suggesting that *mac397* was not dominant negative (Table 1). We propose that *mac397* caused a loss of function.

## CSNK-1 primarily functions in the epidermis to affect the Sisi phenotype and its function is conserved between *C. elegans* and human

We next examined the expression pattern of *csnk-1*. A *GFP* transgene driven by the *csnk-1* endogenous promoter was broadly expressed in the wildtype background (Fig 1C). The expression was obvious in the pharynx, head neurons, epidermis (hypodermis), intestine, ventral cord, dorsal cord, vulval muscles and body-wall muscles (Fig 1D–1H).

To investigate in which tissue *csnk-1* affects the Sisi phenotype, we performed tissue-specific transgene rescue experiments. Driven by an epidermis-specific promoter (*dpy-7p*), *csnk-1* cDNA transgenes strongly rescued the Sisi phenotype of *csnk-1(lf)* mutants (Table 1). However, *csnk-1* transgenes driven by an intestine-specific promoter (*nhx-2p*) failed to rescue (Table 1). Therefore, *csnk-1* likely functions in the epidermis to affect the Sisi phenotype. We previously reported similar results for *tsp-15* [20].

To examine whether the function of *CSNK1G* was conserved, we expressed human *CSNK1G1*, *CSNK1G2* or *CSNK1G3* cDNA transgenes in the epidermis of *csnk-1(lf)* mutants. We found that these transgenes significantly rescued the Sisi phenotype of the mutants and

*CSNK1G2* transgenes appeared to be more effective (Table 1). Hence, the function of *csnk-1* is likely conserved and human *CSNK1G1/2/3* exhibited similar activities.

CSNK1Gs contain a conserved C-terminal palmitoylation signal (TKCCCFFKR) required for membrane localization and the activities of the kinases [34,55]. A conserved palmitoylation sequence was also found in CSNK-1 (VKCCCFRRR, aa 386–394, S1 Fig). We generated two mutant *csnk-1* transgenes disrupting this sequence (p.C388-K407del or p.C388-390S, S1 Fig). These transgenes failed to or barely rescued the Sisi phenotype of *csnk-1(lf)* mutants (Table 1).

## Nonallelic noncomplementation genetic interaction between *csnk-1* and *bli-3/tsp-15/doxa-1*

Since *csnk-1(mac397)* and *tsp-15(mac499)* mutations were both detected in the descendants of the original *mac397* isolate (S1 Table), which was picked as a Sisi $F_1$ progeny of EMS-mutagenized wildtype $P_0$ animals, we speculated that this $F_1$ progeny probably carried a *tsp-15 (mac499) csnk-1(mac397)/+ +* genotype and the rare co-presence of *csnk-1(mac397)/+* and *tsp-15(mac499)/+* might have caused the Sisi phenotype in this animal [20]. [*tsp-15(mac499)* is recessive (S3 Table) so its heterozygous presence would not cause Sisi in our $F_1$ screen.] It is worth noting that we previously observed similar genetic interactions between *bli-3(lf)/+* and *doxa-1(lf)/+* mutations [20]. Such interactions, called nonallelic noncomplementation [56,57], are often exhibited by two genes encoding proteins physically interacting with each other or functioning in the same pathway [56–58]. To examine whether there were broader nonallelic noncomplementation interactions, we generated more double mutants between *csnk-1* and *bli-3/tsp-15/doxa-1*.

We found that *tsp-15(lf) +/+ csnk-1(lf)* males grew like wild type on regular NGM plates (Table 2). On plates with 5 mM NaI, some of these males were weakly Sisi (Table 2). Similarly, *bli-3(lf) csnk-1(lf)/+ +* hermaphrodites exhibited a wildtype-like phenotype on regular NGM plates (Table 2 and S4 and S5 Figs) and some were weakly Sisi in 5 mM NaI (Table 2).

Interestingly, *csnk-1(lf)/+; doxa-1(mac55lf)/+* hermaphrodites were obviously blistered and dumpy on regular NGM plates (Table 2 and S4 and S5 Figs). These animals showed a strong Sisi phenotype (Table 2). A different *doxa-1(lf)* allele (*mac67*) that we previously isolated [20] exhibited similar interactions with *csnk-1(lf)* (Table 2).

Meanwhile, we examined the morphology of double homozygous mutants between *csnk-1 (lf)* and *bli-3(lf)* or *doxa-1(lf)*. Compared to single mutants, *bli-3(lf) csnk-1(lf)* double mutants were strongly scrawny and blistered (S4 and S6 Figs). Differently, *csnk-1(lf); doxa-1(lf)* double homozygotes were apparently larger (S6 Fig) and appeared like *csnk-1(lf)/+; doxa-1(lf)/+* double heterozygotes (S5 Fig).

## Conserved interaction between CSNK-1 and DOXA-1

The genetic findings imply that CSNK-1 and BLI-3/TSP-15/DOXA-1 might form a protein complex or function in the same pathway. Furthermore, the more special interaction between *csnk-1* and *doxa-1* prompted us to speculate whether DOXA-1 might physically interact with CSNK-1. To investigate the possibility, we co-expressed a DOXA-1::GFP fusion protein and a CSNK-1::mCherry fusion protein in *C. elegans* epidermis. Here we detected colocalization of the fluorescent signals in epidermal subcellular structures (Fig 2A–2C). The colocalization is likely caused by specific interaction between DOXA-1 and CSNK-1 because GFP alone did not colocalize with CSNK-1::mCherry, nor did mCherry alone colocalize with DOXA-1::GFP (S7 Fig).

We next transiently expressed a FLAG::CSNK-1 fusion protein with HA tag alone or with a DOXA-1::HA fusion protein in HEK293T cells. From the cell extracts, FLAG::CSNK-1 could

**Table 2. Nonallelic noncomplementation interaction between *csnk-1* and *tsp-15*, *bli-3* or *doxa-1*.**

| Genotype | Phenotype (No NaI) | Survival in 5 mM NaI |
|---|---|---|
| WT | WT | No |
| *csnk-1(mac494lf)/+* | WT-like | No |
| *csnk-1(mac495lf)/+* | WT-like | No |
| *tsp-15(mac500lf)/+* | WT-like (male) | No (male) |
| *tsp-15(mac501lf)/+* | WT-like (male) | No (male) |
| *bli-3(mac40lf)/+* | WT-like (male) | No (male) |
| *bli-3(e767lf)/+* | WT-like (male) | No (male) |
| *doxa-1(mac55lf)/+* | WT-like (male) | No (male) |
| *doxa-1(mac67lf)/+* | WT-like (male) | No (male) |
| *tsp-15(mac500lf) +/+ csnk-1(mac494lf)* | WT-like (male) | Weak (scrawny) (male) |
| *tsp-15(mac500lf) +/+ csnk-1(mac495lf)* | WT-like (male) | Weak (scrawny) (male) |
| *tsp-15(mac501lf) +/+ csnk-1(mac494lf)* | WT-like (male) | Weak (scrawny) (male) |
| *tsp-15(mac501lf) +/+ csnk-1(mac495lf)* | WT-like (male) | Weak (scrawny) (male) |
| *bli-3(mac40lf) csnk-1(mac494lf)/+ +* | WT-like | Weak (scrawny) |
| *bli-3(mac40lf) csnk-1(mac495lf)/+ +* | WT-like | Weak (scrawny) |
| *bli-3(e767lf) csnk-1(mac494lf)/+ +* | WT-like | Weak (scrawny) |
| *bli-3(e767lf) csnk-1(mac495lf)/+ +* | WT-like | Weak (scrawny) |
| *csnk-1(mac494lf)/+; doxa-1(mac55lf)/+* | Bli, Dpy | Strong (Bli, Dpy) |
| *csnk-1(mac495lf)/+; doxa-1(mac55lf)/+* | Bli, Dpy | Strong (Bli, Dpy) |
| *csnk-1(mac494lf)/+; doxa-1(mac67lf)/+* | Bli (male) | Strong (Bli) (male) |
| *csnk-1(mac495lf)/+; doxa-1(mac67lf)/+* | Bli (male) | Strong (Bli) (male) |

*tsp-15(mac500lf)* and *tsp-15(mac501lf)* mutations were knock-in genocopies of *tsp-15(mac499lf)* (S3 Table). *csnk-1 (lf)/hT2* was treated as *csnk-1(lf)/+*. We failed to place *tsp-15(lf)* and *csnk-1(lf)* on the same chromosome due to the close linkage of the two genes. For this reason, we only examined male progeny from crosses between *tsp-15(lf)* males and *csnk-1(lf)/hT2* hermaphrodites. *bli-3(lf) csnk-1(lf)/hT2* and *csnk-1(lf)/hT2; doxa-1(mac55lf)/hT2* heterozygotes were treated as *bli-3(lf) csnk-1(lf)/+ +* and *csnk-1(lf)/+; doxa-1(lf)/+* mutants. *csnk-1(lf)/+; doxa-1(mac67lf)/+* double heterozygous males were generated by crossing *doxa-1(mac67lf)* males with *csnk-1(lf)/hT2* hermaphrodites.

be coimmunoprecipitated with DOXA-1::HA using an anti-HA antibody, while FLAG:: CSNK-1 co-expressed with HA tag alone could not be coimmunoprecipitated using this anti-body (Fig 2D). We further performed reverse immunoprecipitation using an anti-FLAG anti-body. Similarly, DOXA-1::HA could be coimmunoprecipitated with FLAG::CSNK-1 but not with the FLAG tag alone (Fig 2E).

To examine the interaction between CSNK-1 and DOXA-1 by a different approach, we purified a bacterially expressed His::TF::DOXA-1 fusion protein and co-incubated the purified protein with the lysates of HEK293T cells expressing FLAG::CSNK-1. Here FLAG::CSNK-1 could be specifically pulled down by His::TF::DOXA-1 (Fig 2F). In reverse, a His::TF::CSNK-1 fusion protein purified from bacterial expression could specifically pull down a DOXA-1::HA fusion protein expressed in HEK293T cells (Fig 2G). Together, these results suggest a specific interaction between CSNK-1 and DOXA-1. Nevertheless, caution should be exercised in inter-preting the pulldown results as proteins expressed in *E. coli* may be incorrectly folded, which might cause non-specific interactions.

DOXA-1 is orthologous to the human DUOX maturation factor 1 and 2 (DUOXA1 and DUOXA2) (S8 Fig), which are required for dual oxidase activities [17,59]. To investigate whether human CSNK1Gs and DUOXAs exhibit similar interactions like that between CSNK-

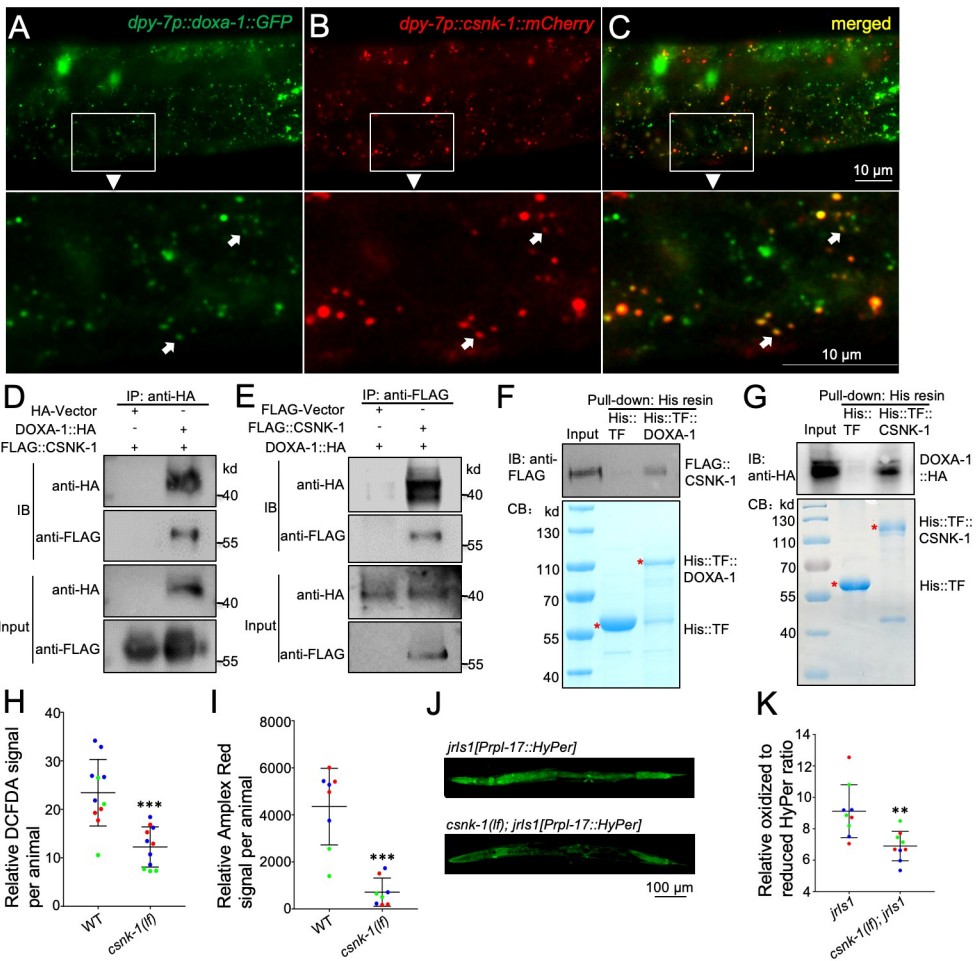

**Fig 2. CSNK-1 interacts with DOXA-1.** (A, B, C) Confocal pictures of a transgenic L3 larva co-expressing *dpy-7p*::*doxa-1*::*GFP* and *dpy-7p*::*csnk-1*::*mCherry* transgenes. Enclosed areas are enlarged in lower panels. Arrows indicate typical subcellular structures co-labeled by GFP and mCherry. (D, E) Western blotting showing the specific interaction between DOXA-1::HA and FLAG::CSNK-1 expressed in HEK293T Cells. Immunoprecipitation was performed using an anti-HA antibody (D) or an anti-FLAG antibody (E). (F, G) Western blotting showing that His::TF::DOXA-1 or His::TF::CSNK-1 purified from BL21 bacteria can specifically pull-down FLAG::CSNK-1 (F) or DOXA-1::HA (G) expressed in HEK293T Cells, respectively. Coomassie blue (CB) staining of purified proteins is shown in lower panels. (H, I) Relative fluorescent intensities of L4 larva stained with DCFDA or Amplex Red. Results were based on three biological replicates, with 2–5 samples per replicate. Each datapoint represents one sample. Colors represent different replicates. Statistics: two-tailed unpaired Student's *t*-test. ***: $p < 0.001$. (J) Representative fluorescent pictures of the *jrIs1* reporter in wildtype or *csnk-1(lf)* L4 larva. Picture exposure time was 432.6 ms. (K) Relative oxidized/reduced HyPer reporter signals. Results were based on three biological replicates, with 2–3 samples per replicate. Each data point represents one sample. Colors represent different replicates. Statistics: two-tailed unpaired Student's *t*-test. **: $p < 0.01$. Error bars: standard deviation.

1 and DOXA-1, we co-expressed FLAG::CSNK1G2 and DUOXA2::HA fusion proteins in HEK293T cells. In these cells we detected co-localization of the two proteins on the plasma membrane and subcellular structures (Fig 3A). Similar colocalization was also observed in a transfected HeLa cell (S3I Fig). [CSNK1G2 was chosen because its transgenes showed stronger rescuing effects (Table 1).] We also detected a potentially specific interaction between these proteins using immunoprecipitation (Fig 3B).

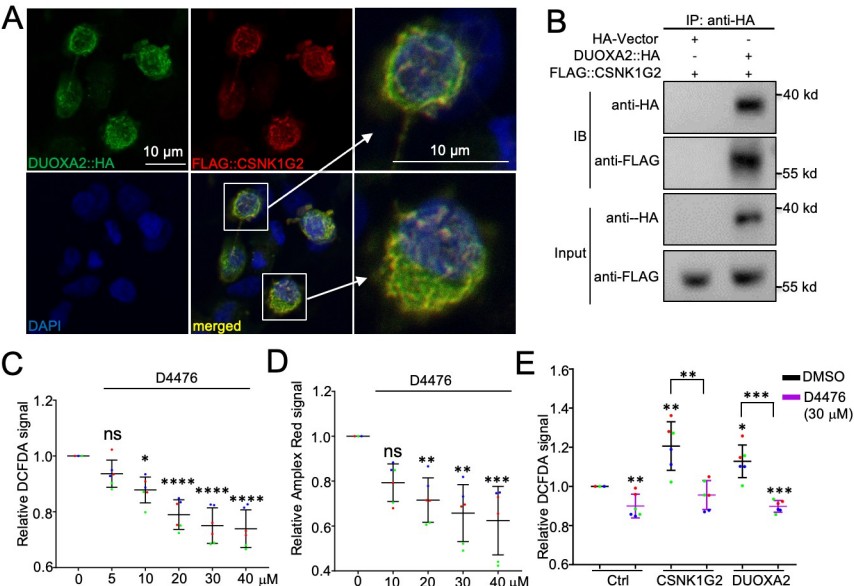

**Fig 3. Human CSNK1G2 interacts with DUOXA2 and promotes ROS levels.** (A) Immunofluorescent staining showing membrane and subcellular colocalization of DUOXA2::HA (green) and FLAG::CSNK1G2 (red) co-expressed in HEK293T cells. Enlarged pictures of two cells are shown on right. We observed that CSNK1G2 overexpression often changed the fibroblast morphology of the cells to a round shape. (B) Western blotting showing the interaction between DUOXA2::HA and FLAG::CSNK1G2 co-expressed in HEK293T Cells. Immunoprecipitation was performed using an anti-HA antibody. (C, D) Dose-dependent inhibitory effect of D4476 on DCFDA or Amplex Red signals in HEK293T cells. Three biological replicates were performed with two wells per replicate. Colors represent different replicates. Statistics: Bonferroni's multiple comparison test with one-way ANOVA. *: $p < 0.05$; **: $p < 0.01$; ***: $p < 0.001$; ****: $p < 0.0001$. ns: not significant. (E) Effects of D4476 on DCFDA signals in HEK293T cells overexpressing CSNK1G2 or DUOXA2. Controls were transfected with pCMV-Tag2B backbone vector. Three biological replicates were performed with two wells per replicate. Colors represent different replicates. See raw data for all results. Statistics: two-tailed unpaired Student's $t$-test. *: $p < 0.05$; **: $p < 0.01$; ***: $p < 0.001$.

## CSNK-1 promotes ROS levels

The interaction between CSNK-1 and DOXA-1 suggests that CSNK-1 might affect ROS levels. Using 2',7'-dichlorofluorescin diacetate (DCFDA) and Amplex Red staining as readouts of ROS levels, we found that *csnk-1(lf)* mutants exhibited significantly reduced ROS levels (Figs 2H, 2I, S3J and S3K). We further crossed the *jrIs1* reporter [60], an YFP-based hydrogen peroxide sensor that detects endogenous ROS levels, into *csnk-1(lf)* mutants and observed obviously reduced levels of oxidized sensor proteins in the mutants (Fig 2J and 2K).

To investigate whether human CSNK1 might affect ROS levels, we treated HEK293T cells with D4476, a selective small molecule inhibitor of CSNK1 [61]. Using DCFDA and Amplex Red staining, we detected a dose-dependent inhibition of ROS levels by D4476 (Fig 3C and 3D). Furthermore, overexpression of CSNK1G2 led to increased ROS levels in the cells as measured by DCFDA fluorescence, which were inhibited by D4476 (Fig 3E). The increased ROS levels caused by DUOXA2 overexpression were also inhibited by D4476 (Fig 3E).

The interaction between CSNK-1 and DOXA-1 prompted us to explore whether DOXA-1 activity might be related to any potential phosphorylation sites. We inspected DOXA-1 sequence using Scansite 4 (www.scansite4.mit.edu) and the top three predicted sites were shown in S8 Fig (red boxes and red arrowheads). These sites have the characters of the consensus CSNK-1 phosphorylation sites, D/E/(p)S/T(X)$_{1-3}$S/T [55], in which the phospho-acceptor residue is underlined. We mutated each individual residue to alanine (A) in *doxa-1* transgenes.

**Table 3. Effects of predicted potential casein kinase 1 phosphorylation sites on DOXA-1 activity.**

| Background | *doxa-1p:: doxa-1_cDNA Tg* | Tg line | Survival rate in 5 mM NaI (%) (n) | | | | |
|---|---|---|---|---|---|---|---|
| | | | Experiment 1 | Experiment 2 | Experiment 3 | Average | *p*-value (vs *doxa-1(mac55lf)*) |
| *doxa-1 (mac55lf)* | No *Tg* | - | 78.64% (543) | 79.39% (655) | 82.91% (714) | 80.31% | - |
| | *WT Tg* | 1 | 19.55% (266) | 20.91% (330) | 27.84% (255) | 22.77% | 3.736E-05 **** |
| | | 2 | 28.03% (892) | 31.27% (403) | 31.82% (751) | 30.37% | 9.42E-06 **** |
| | | 3 | 33.69% (282) | 33.63% (568) | 32.98% (476) | 33.43% | 3.949E-06 **** |
| | *p.T343A Tg* | 1 | 62.00% (429) | 73.18% (466) | 60.34% (532) | 65.17% | 0.0233645 * |
| | | 2 | 73.13% (454) | 59.31% (693) | 65.77% (298) | 66.07% | 0.0275802 * |
| | | 3 | 59.28% (415) | 73.70% (365) | 72.33% (571) | 68.43% | 0.0678938 |
| | | 4 | 64.24% (302) | 60.85% (447) | 67.92% (293) | 64.34% | 0.002767 ** |
| | *p.S346A Tg* | 1 | 9.25% (227) | 13.14% (274) | 12.73% (330) | 11.71% | 2.86E-06 **** |
| | | 2 | 18.52% (297) | 14.17% (247) | 21.08% (223) | 17.92% | 1.319E-05 **** |
| | | 3 | 9.01% (355) | 13.02% (407) | 12.17% (337) | 11.40% | 2.75E-06 **** |
| | *p.S364A Tg* | 1 | 17.33% (329) | 15.12% (258) | 13.77% (305) | 15.40% | 2.659E-06 **** |
| | | 2 | 8.42% (190) | 8.97% (223) | 6.36% (283) | 7.92% | 1.221E-06 **** |
| | | 3 | 8.25% (194) | 17.41% (201) | 17.20% (186) | 14.29% | 3.665E-05 **** |

*doxa-1(mac55lf)* mutants carrying the indicated *doxa-1* transgenes were observed for the Sisi phenotype in 5 mM NaI. n: total number of L1 larva examined. Statistics: two-tailed unpaired Student's *t*-test.

Interestingly, only the T343A mutation significantly weakened the rescuing effects of the transgene (Table 3). These results suggest that T343 is important for DOXA-1 activity. However, it remains unclear whether the importance of this site depends on its phosphorylation.

## *csnk-1* and *skn-1* genetically interact to affect oxidative stress response

*skn-1(gf)* mutants are Sisi while *skn-1(lf)* mutants are not [20]. To understand the interaction between *csnk-1* and *skn-1*, we examined the Sisi phenotype of *csnk-1; skn-1* double mutants.

Different from *skn-1(lf)* mutants, which are not Sisi, *csnk-1(lf); skn-1(lf)* double mutants became Sisi (Table 4). This implies an epistatic effect of *csnk-1* on the Sisi phenotype. We next examined the Sisi phenotype of *csnk-1(lf); skn-1(gf)* double mutants in 50 mM NaI, a test that we previously used to examine whether *bli-3/tsp-15/doxa-1* loss-of-function mutations and *skn*-1 gain-of-function mutations had synergistic effects [20]. Though single mutant failed to show the Sisi phenotype in such a high concentration of NaI, *csnk-1(lf); skn-1(gf)* double mutants became Sisi (Table 5).

Finally, to investigate whether CSNK-1 might affect the expression of genes regulated by SKN-1, we crossed a SKN-1 target gene reporter, *dvIs19 (gst-4p::GFP)* [62] into *csnk-1(lf)* mutants. The GFP signals were obviously increased in both *csnk-1(lf)* mutants (S9A Fig) and the increase was still visible when the animals were treated with *skn-1(RNAi)* (S9B Fig).

## Discussion

In this study, we identified a novel role of CSNK-1 CSNK1G in oxidative stress response. We uncovered nonallelic noncomplementation interactions between *C. elegans csnk-1* and *bli-3/tsp-15/doxa-1* NADPH dual oxidase genes, which led to the findings of a potentially conserved biochemical interaction between CSNK-1 and DOXA-1 and a conserved function of CSNK-1 in promoting ROS levels. We also detected genetic interactions between *csnk-1* and *skn-1* in oxidative stress response.

**Table 4. Sisi phenotype of animals carrying different combinations of *csnk-1(lf)* and *skn-1(zu135lf)* mutations.**

| $P_0$ genotype | $F_1$ genotype | $F_1$ adult number | | | |
|---|---|---|---|---|---|
| | | 0 mM NaI | | 5 mM NaI | |
| *csnk-1(mac494lf)/+; skn-1(zu135lf)/+* | *+/+; +/+* | ND | | 0 | |
| | *+/+; skn-1(lf)/+* | ND | | 0 | |
| | *csnk-1(mac494lf)/+; +/+* | ND | | 0 | |
| | *csnk-1(mac494lf)/+; skn-1(lf)/+* | ND | | 0 | |
| | *csnk-1(mac494lf)/+; skn-1(lf)* | 15 | | 0 | |
| | *csnk-1(mac494lf); +/+* | 7 | | 61 | |
| | *csnk-1(mac494lf); skn-1(lf)/+* | 14 | | 136 | |
| | *csnk-1(mac494lf); skn-1(lf)* | 3 | $p = 0.10557$ | 29 | $p = 0.00231$ ** |
| | *+/+; skn-1(lf)* | 7 | | 0 | |
| *csnk-1(mac495lf)/+; skn-1(zu135lf)/+* | *+/+; +/+* | ND | | 0 | |
| | *+/+; skn-1(lf)/+* | ND | | 0 | |
| | *csnk-1(mac495lf)/+; +/+* | ND | | 0 | |
| | *csnk-1(mac495lf)/+; skn-1(lf)/+* | ND | | 0 | |
| | *csnk-1(mac495lf)/+; skn-1(lf)* | 15 | | 0 | |
| | *csnk-1(mac495lf); +/+* | 4 | | 6 | |
| | *csnk-1(mac495lf); skn-1(lf)/+* | 10 | | 27 | |
| | *csnk-1(mac495lf); skn-1(lf)* | 1 | $p = 0.34905$ | 10 | $p = 0.00299$ ** |
| | *+/+; skn-1(lf)* | 6 | | 0 | |

$F_1$ progeny of *csnk-1(lf)/+; skn-1(zu135lf)/+* $P_0$ animals were grown on plates with or without 5 mM NaI. For plates without NaI, $F_1$ adults were randomly picked to separate plates and examined for egg laying and hatching. Individuals that laid hatched eggs were not analyzed further as they were not homozygous at either *csnk-1* or *skn-1* locus. Individuals that only laid unhatched eggs were genotyped for *csnk-1(lf)* and *skn-1(lf)* mutations. For plates with 5 mM NaI, $F_1$ adults were individually genotyped for *csnk-1(lf)* and *skn-1(lf)* mutations. ND: genotypes not determined. Statistics: two-tailed unpaired Student's *t*-test. Comparison was made between *csnk-1(lf); skn-1(lf)* and *+/+; skn-1(lf)*.

## CSNK-1 is an embryonically essential kinase with novel function in postembryonic oxidative stress response

*csnk-1* is an essential gene that affects the embryonic asymmetric spindle positioning and oocyte meiosis in *C. elegans* [44–49]. Consistently, we found that eggs laid by *csnk-1(lf)* mutants failed to hatch. The effect of *csnk-1* on the Sisi phenotype is specific, as RNAi knockdown of *kin-19*, *kin-20* and three other kinase genes did not obviously affect the phenotype

**Table 5. *csnk-1(lf)* and *skn-1(gf)* mutations exhibit synergy on the Sisi phenotype in 50 mM NaI.**

| Genotype | Survival rate in 50 mM NaI (%) (n) | | | | |
|---|---|---|---|---|---|
| | Experiment 1 | Experiment 2 | Experiment 3 | Average | *p*-value (vs *csnk-1(lf)* or *skn-1(gf)*) |
| *csnk-1(mac494lf)* | 0% (246) | 0% (295) | 0% (255) | 0.00% | - |
| *csnk-1(mac495lf)* | 0% (182) | 0% (137) | 0% (230) | 0.00% | |
| *skn-1(lax120gf)* | 0% (285) | 0% (279) | 0% (327) | 0.00% | - |
| *skn-1(mac53gf)* | 0% (335) | 0% (397) | 0% (310) | 0.00% | |
| *csnk-1(mac494lf); skn-1(lax120gf)* | 69.55% (220) | 81.28% (235) | 77.82% (284) | 76.21% | 0.00208 ** |
| *csnk-1(mac495lf); skn-1(lax120gf)* | 40.10% (197) | 39.90% (208) | 37.67% (223) | 39.22% | 0.00040 *** |
| *csnk-1(mac494lf); skn-1(mac53gf)* | 59.78% (276) | 64.17% (307) | 70.12% (405) | 64.69% | 0.00214 ** |
| *csnk-1(mac495lf); skn-1(mac53gf)* | 33.06% (242) | 35.91% (181) | 32.68% (205) | 33.88% | 0.00090 *** |

n: total number of L1 larva examined. Statistics: two-tailed unpaired Student's *t*-test.

(S2 Table). The importance of the conserved C-terminal palmitoylation for CSNK-1 activity is supported by the finding that mutations of the site caused a loss of the function.

Our transgene rescue results suggest functional similarity among the three human CSNK1Gs in affecting oxidative stress response. This is consistent with a recent finding that human CSNK1Gs exhibited functional redundancy in regulating Wnt signaling [63]. Together with our findings that CSNK-1 and human CSNK1G2 can promote ROS levels, we suggest that CSNK1Gs are important regulators of ROS homeostasis across species.

## CSNK1G interacts with the NADPH dual oxidase complex

Mammals have seven membrane-localized NADPH oxidases as major generators of endogenous ROS [2,31], among which the dual oxidase DUOX2 is essential for thyroid hormone synthesis, while DUOX1 is more involved in immune cell functions [31]. It is not well understood how DUOX activities are regulated. DUOXA1 and DUOXA2 are conserved facilitators for the ER-Golgi-plasma membrane translocation of DUOX1 and DUOX2 [59]. In *C. elegans*, BLI-3 DUOX, DOXA-1 DUOXA and TSP-15 tetraspanin form a dual oxidase complex [16,17]. Recently, the highly conserved MEMO-1 protein was identified as a negative regulator of BLI-3 activity via binding the RHO-1/RhoA/GTPase [15], providing a new regulatory mechanism for dual oxidase activities.

We suggest that CSNK-1 interacts with the dual oxidase complex and might regulate dual oxidase activity. First, the nonallelic noncomplementation interactions between *csnk-1* and *bli-3/tsp-15/doxa-1* imply physical and/or functional interactions between CSNK-1 and BLI-3/TSP-15/DOXA-1. Second, CSNK-1 was colocalized with DOXA-1 on *C. elegans* epithelial subcellular structures. So did human CSNK1G2 and DUOXA2 in HEK293T cells. Third, that CSNK-1 and DOXA-1 were detected in a same protein complex by immunoprecipitation and pull-down experiments suggests a possible direct physical interaction between CSNK-1 and DOXA-1, and human CSNK1G2 and DUOXA2 were detected in a same protein complex by immunoprecipitation. Fourth, CSNK-1 is required for normal ROS levels in *C. elegans* and CSNK1G2 and DUOXA2 each can promote ROS levels in HEK293T cells.

We found that D4476, a commonly used CSNK1 inhibitor, can decrease ROS levels in HEK293T cells in a dose-dependent manner and suppress the promotion of ROS levels by overexpressed CSNK1G2 and DUOXA2, supporting the involvement of CSNK1 in regulating ROS levels. However, our evidence is insufficient for connecting CSNK1s directly with ROS level regulation before extensive analyses of the expression and function of each CSNK1 in HEK293T are performed.

Mutagenesis analyses suggest that T343, the phospho-acceptor of a potential CSNK1 site in DOXA-1 C-terminal domain (S8 Fig), was required for DOXA-1 activity, implying phosphorylation as a mechanism for regulating DOXA-1 activity. However, the lack of conservation in the aligned DUOXA2 sequence raises the question about the importance of this site in other species. Considering that four potential CSNK1 phosphorylation sites are detected in the C-terminal domain of DUOXA2 (S8 Fig), two of which appear to be conserved in DOXA-1, we suspect that the interaction between CSNK-1 and DOXA-1 and the phosphorylation of DOXA-1, if it exists, might be located at different sites. To answer this question, it is necessary to identify the interacting domains between these proteins, the potential phosphorylation sites on DOXA-1, and the functional importance of these sites in future.

### *csnk-1* genetically interacts with *skn-1*

SKN-1 is the *C. elegans* ortholog of Nrf2 [8,9] and a key regulator of oxidative stress response and other cellular processes [6]. We previously found that *skn-1* can genetically interact with

*bli-3/tsp-15/doxa-1* to affect the Sisi phenotype [20]. Similarly in this study we found that *csnk-1* and *skn-1* can genetically interact to affect the Sisi phenotype. Specifically, the lack of Sisi phenotype in *skn-1(lf)* mutants in 5 mM NaI was suppressed by *csnk-1(lf)*, while *csnk-1(lf)* and *skn-1(gf)* exhibited synergy in promoting the Sisi phenotype in 50 mM NaI. These interactions imply an opposite effect of *csnk-1* on oxidative stress response that is in parallel with *skn-1*. However, the molecular genetic mechanism underlying such interactions remains to be understood considering that the genetic analyses were performed on mutants with maternal effects and that reducing ROS levels (like in *bli-3/tsp-15/doxa-1* loss-of-function mutants) [19,20], increasing antioxidant expression (like in *skn-1(gf)* mutants) [20] or adding antioxidants such as vitamin C or NAC to NGM plate [20] all can cause the Sisi phenotype.

We found that the expression of the SKN-1 reporter transgene *dvIs19* (*gst-4p::GFP*) was increased in *csnk-1(lf)* mutants and the increase existed even with *skn-1(RNAi)* treatment. However, considering that RNAi knockdown of *skn-1* could be partial, we have yet to interpret such an effect of *csnk-1* as *skn-1*-independent. To gain a deeper understanding of the interaction between CSNK-1 and SKN-1, it will be important to determine how ROS levels are affected in *csnk-1; skn-1* double mutants, whether endogenous *gst-4* expression is similarly affected by *csnk-1(lf)*, or whether any other gene expression is affected by *csnk-1*. We hope to address these questions in future using multidisciplinary approaches such as transcriptome and phosphoproteomic analyses.

## Conclusions

Among casein kinase 1 members, CSNK1Gs are understudied and are classified by the NIH program "Illuminating the Druggable Genome" as "dark kinases" calling for extra exploration (https://commonfund.nih.gov/IDG). We uncovered an important role of CSNK-1 CSNK1G in regulating oxidative stress response and ROS levels, which involves potentially conserved interactions between CSNK1G and the NADPH dual oxidase complex. Our findings provide novel insights into redox biology and may facilitate the understanding of diseases caused by mutations in these genes and more broadly by defective oxidative stress response.

## Materials and methods

### Mapping and cloning of *csnk-1*

*mac397* was previously isolated as a Sisi $F_1$ progeny of EMS-mutagenized wildtype $P_0$ animals grown on plates with 5 mM NaI [20]. To map *mac397*, males of the Hawaiian strain CB4856 were crossed with *mac397* hermaphrodites. $F_1$ male progeny were crossed with CB4856 hermaphrodites on plates with 5 mM NaI to generate Sisi $F_2$ male progeny. Similar crosses were performed three more rounds to generate Sisi $F_5$ hermaphrodites, which were picked to individual plates with 5 mM NaI and allowed to propagate. Animals on these plates were examined for single nucleotide polymorphisms (SNPs) [64,65]. We established a linkage on Chr. I to the right of SNP *WBVar00240395* (SNP W03D8: 34384 D1 = T, genetic location: -5.68; GenBank accession no. FO081764) and to the left of SNP *WBVar00245221* (SNP F58D5: 17029–17032 D4 = ACTA, genetic location: 13.02; GenBank accession no. AL137227).

To determine the genomic sequences of *mac397* mutants, progeny of unbackcrossed original isolate were washed and starved for ~4 hrs in $H_2O$. Genomic DNAs were extracted by proteinase K digestion, followed by RNase A treatment and two rounds of phenol–chloroform extraction. Three genomic DNA libraries (350 bp inserts) were constructed by Annoroad Gene Technology Corporation (Beijing) using Illumina's paired-end protocol. Paired-end sequencing (100-bp reads) was performed on the Illumina HiSeq X Ten. 4G clean bases were mapped to the N2 genome (Wormbase release 220) after removal of duplicated reads.

Genomic sequencing identified 7 homozygous and 4 heterozygous missense mutations between the mapped SNPs on Chr. I, among which were a homozygous mutation in *tsp-15* and a heterozygous mutation in *csnk-1* (S1 Table). The homozygosity of the *tsp-15* mutation might be caused by prolonged propagation of the original isolate in 5 mM NaI before the strain was sequenced (see below).

We initially suspected that the *tsp-15* mutation (exon3:c.G331A:p.G111R, named *mac499*) (S1 and S3 Tables) might be a dominant negative allele, thus causing the Sisi phenotype of the *mac397* isolate as a heterozygote. To test this, we generated two knock-in mutations genocopying *mac499* (S3 Table, *mac500* and *mac501*). The heterozygous knock-in mutants were not Sisi, while the homozygous mutants were (S3 Table). In addition, a *tsp-15* transgene can rescue the Sisi phenotype of *tsp-15(mac499)* homozygous mutants (S3 Table). Therefore, *tsp-15 (mac499)* is recessive, which led us to suspect that a linked mutation, either alone as heterozygote or together with *tsp-15(mac499)/+*, caused the Sisi phenotype of the original isolate.

To identify this gene, we treated wildtype animals with feeding RNAi targeting the genes listed in S1 Table. Besides *tsp-15(RNAi)*, we found that *csnk-1(RNAi)* also made animals Sisi (S1 and S2 Tables). The corresponding mutation in *csnk-1* was name *mac397*.

After we found that *csnk-1(RNAi)* can cause Sisi, we re-examined our *mac397* frozen stocks but were only able to detect *tsp-15(mac499)* mutation. *mac397*, either as heterozygote or homozygote, was not found in different batches of thawed animals.

Considering that *mac397* was a loss-of-function mutation (Table 1) and *csnk-1(lf)* homozygous mutations cause lethality, we postulate that a prolonged propagation of the mutants before freezing probably caused this phenomenon. In the original isolate, *csnk-1(mac397)* was likely heterozygous (S1 Table, genomic sequencing results in the early propagation phase). A prolonged propagation of the strain in 5 mM NaI, with *tsp-15(mac499) csnk-1(mac397)/+ +* as the presumptive starting genotype, would generate crossover between *tsp-15* and *csnk-1* and allow *tsp-15(mac499)* (viable as heterozygotes or homozygotes) to outcompete *csnk-1(mac397)* (only viable as heterozygotes) in the population. At the time of freezing, most animals, if not all, might have carried only a *tsp-15(mac499)* homozygous mutation and lost the *csnk-1 (mac397)* mutation.

We previously found that *bli-3(lf)/+; doxa(lf)/+* double heterozygous mutants were Sisi in 5 mM NaI, while *bli-3(lf)/+* or *doxa-1(lf)/+* single mutants were not [19,20], suggesting a non-allelic noncomplementation interaction between the two genes. Similarly, we postulate that *csnk-1(mac397)* and *tsp-15(mac499)* probably interacted by nonallelic noncomplementation to cause the Sisi phenotype in the original *mac397* isolate [20].

### *C. elegans* survival assay in excess iodide (Sisi phenotype assay)

The survival assay was performed as described [19] with modification.

In general, bleached eggs were grown on OP50-seeded NGM plates with different concentrations of NaI (5 mM or 50 mM). The number of L1 larva was counted and the number of adults was counted 3 days post L1.

For heterozygous males in Table 2, males and hermaphrodites of the desired genotypes were crossed on plates with 5 mM NaI. The cross progeny were allowed to grow in the same plate for 4–6 days and the appearance of young adult males was recorded.

### Generation of *csnk-1* or *tsp-15* mutations using CRISPR/Cas9

We followed the method [66] with modifications. The DNA mixture for injection contained 50 ng/μl *Peft-3::Cas9-SV40_NLS*, 25 ng/μl *PU6::sgRNA* (specific for *csnk-1* or *tsp-15*), and 25 ng/μl *PU6::sgRNA* (specific for *dpy-10*) as co-injection marker. To generate *tsp-15* knockin

mutations, we included a 130 nt synthesized oligo (500 nM) containing the desired *tsp-15* mutation as the repair template and a repair template for *dpy-10* as described [67]. $F_1$ animals with the Dpy/Rol phenotype were picked to individual plates and their progeny were analyzed for the desired mutations by Sanger sequencing. Target sequences of sgRNAs and the sequence of *tsp-15* repair template are shown in S6 Table.

*mac494* and *mac495* homozygous mutants derived from heterozygous parents can grow into adults and were able to lay multiple eggs. However, these eggs would not or rarely hatch. We maintained *csnk-1* mutations using the *hT2 [bli-4(e937) let-?(q782) qIs48] (I; III)* balancer.

## Transgene experiments

Germline transgene experiments were performed as described [68].

For *csnk-1* rescue experiments, the transgene mixture containing 0.5 ng/µl or 5 ng/µl of the transgenes of interest with 20 ng/µl pPD95_86 (*myo-3p*::*GFP*) (a gift from Andrew Fire) as co-injection marker was injected to wildtype animals to generate at least two stable lines. The transgenes were crossed into *csnk-1(mac494lf)/hT2[qIs48]* animals and *csnk-1(mac494lf)* homozygous progeny carrying the transgenes were examined. For the Sisi phenotyping, transgenic adults were bleached, and eggs were placed on regular or 5 mM NaI NGM plates for 12 hrs to hatch. ~100 transgenic homozygous *csnk-1(mac494lf)* L1 larva were picked to a new 5 mM NaI NGM plates. The numbers of adults were counted after 3 days.

## Hoechst 33258 staining

Hoechst 33258 staining was performed as described [16,19] with modifications. Young adult animals (24 hrs post mid-L4) were washed off plates and incubated for 15 min in M9 containing 1 µg/ml Hoechst 33258 (Sigma, 861405) at 20˚C with gentle shaking. After staining, animals were washed three times with M9 and observed under a Leica DM5000B fluorescence microscope.

## RNA interference

Bleached eggs were transferred to plates seeded with HT115 (DE3) bacteria expressing RNAi plasmids on NGM plates with 1 mM IPTG and 0.1 mg/ml Ampicillin. The progeny were examined under dissecting microscope for the Sisi phenotype. RNAi feeding bacterial strains for *lrp-2*, *rbpl-1*, *uggt-2*, *K07A1.1*, *sys-1 and gsk-3* were picked from a whole-genome RNAi library [69], and the inserts were confirmed by sequencing. RNAi plasmids for other genes were generated in this study (see Plasmids).

## Confocal microscopy

Fluorescent images of immunostained cells or transgenic animals expressing GFP and/or mCherry reporters were captured using a Leica TCS SP5 II or Zeiss LSM 880 laser confocal microscope.

## Cell culture and transfection

HeLa and HEK293T cells were cultured in DMEM medium supplemented with 10% fetal bovine serum (Gibco, 12484028) and 1% penicillin/streptomycin (ThermoFisher Scientific,15140163). Cells were maintained at 37˚C with 5% $CO_2$. Plasmid transfection was performed at 80% cell confluency using Lipofectamine 3000 Reagent (Invitrogen, L3000015) according to the manufacturer's instructions. Cells were analyzed after 48 hrs.

## Immunostaining

For immunostaining, HEK293T or HeLa cells grown on coverslips were fixed with 4% PFA in PBS at room temperature for 15 min and permeabilized with 0.25% PBST (PBS containing 0.25% Triton X–100) at room temperature for 15 min. Cells were blocked in 5% normal goat serum for 1 hr at room temperature. The samples were incubated overnight at 4˚C with primary antibodies: rabbit anti-HA-tag (1:500; Cell Signaling Technology, #3724) and mouse anti-FLAG-tag (1:500; Sigma, F9291). Cells were washed in PBST and Alexa Fluor 488-anti-rabbit (1:200; Jackson Immunoresearch, 111545144) and Cy3-anti-mouse (1:200; Jackson Immunoresearch, 115165003) were used as the secondary antibodies. The samples were then stained with DAPI (1 μg/mL, Sigma, D9542) for 5 min at room temperature, washed, and mounted with Fluoromount aqueous mounting medium (Sigma, F4680).

## Immunoprecipitation, pull-down and western blotting

For immunoprecipitation, HEK293T cells were lysed in ice-cold IP buffer (Beyotime, P0013) with 1% protease inhibitor cocktail and phosphatase inhibitor cocktail (Bimake, B14001). Lysed cells were incubated at 4˚C for 4 hrs and centrifuged at 12000 rpm for 20 min at 4˚C. The clear supernatants were transferred to a new tube and incubated with an anti-HA antibody (1 μl per sample, Cell Signaling Technology, #3724) or anti-FLAG antibody (1 μl per sample, Sigma, F9291) and protein A/G-beads (Bimake, B23201, 10 μl) at 4˚C overnight. The beads were centrifuged and washed 5 times in TBST. Proteins associated with the beads were mixed with 32 μl 2x SDS lysis buffer and 8 μl 5x SDS loading buffer and boiled for 5 min at 95˚C.

For pull-down, BL21 bacteria transformed with *pCold TF*:: *doxa-1_cDNA* or *pCold TF*:: *csnk-1_cDNA* plasmid were cultured in Luria broth (LB) at 37˚C to OD 0.6, and 1 mM IPTG was added to induce protein expression at 16˚C for 24 hrs. Bacteria were collected and lysed in PBST (1% Triton X-100) with 1 mM lysozyme (Beyotime, ST206) on ice for 30 min, followed by centrifugation at 12000 rpm, 4˚C for 20 min. The clear supernatant was transferred to a new tube and incubated with His-tag resin (Beyotime, P2229S-1) at 4˚C for 2 hrs. The resin was collected after centrifugation at 1200 rpm for 20 sec and used to incubate at 4˚C overnight with the lysate supernatant of HEK293T cells transfected with *pCMV-Tag2B (FLAG)*::*csnk-1_cDNA* or *pcDNA3.1-doxa-1_cDNA*::*HA* plasmid. On the next day the mixture was washed three times in TBST. Proteins associated with the resin were mixed with 20 μl 2x SDS lysis buffer and 5 μl 5x SDS loading buffer and boiled for 5 min at 95˚C.

Proteins were separated by SDS-PAGE and transferred to polyvinylidene fluoride (PVDF) membranes (Immobilon-P, IPVH00010). The membranes were incubated with primary antibodies (rabbit anti-HA-tag, 1:5000; Cell Signaling Technology, #3724 or mouse anti-FLAG-tag, 1:5000; Sigma, F9291) at 4˚C overnight. On the next day, the membranes were washed three times in TBST at room temperature, incubated with secondary antibodies (1:10000; Jackson Immunoresearch, anti-mouse 115035146 or anti-rabbit 111035144) at room temperature for 1 hr. Protein bands were visualized using an ECL detection system (YEASEN, 36222ES60).

## ROS measurement using DCFDA

Whole-animal ROS levels were measured using 2,7-dichlorofluorescein diacetate (DCFDA) (Sigma, D6883) following previous methods [70,71] with modifications. Synchronized L4 larval animals were washed with $H_2O$ for three times. The population density was adjusted with $H_2O$ to 50 to 200 animals per 100 μl. Two to three 100 μl aliquots were transferred to individual wells of a black-walled 96-well plate. DCFDA dissolved in DMSO was added to a final concentration of 50 μM. Animals were incubated at 20˚C for 30 min and measured for DCFDA

fluorescence intensity using a fluorimeter (BioTek, Synergy 2) at the excitation wavelength of 485 nm and the emission wavelength of 528 nm at room temperature. After the measurement, animals in each well were emptied to an agar plate and counted.

For DCFDA signals of individual animals, synchronized L4 animals were washed three times in $H_2O$ and resuspended in M9 to a final volume of 95 μl. 5 μl 1 mM DCFDA solution was added to a final concentration of 50 μM. Animals were incubated at 20˚C for 30 min, washed in M9 three times, transferred to a glass slide, and covered with a cover glass. Fluorescent pictures of multiple animals were taken immediately with the same exposure intervals. Whole body fluorescence intensities of individuals were measured with ImageJ.

To measure dose-dependent effects of D4476 (Sigma, D4476) on ROS levels, HEK293T cells at 70% confluency in 24-well plates (coated with 0.01 mg/ml poly-D-lysine (Sigma, P6407)) were treated with different concentrations of D4476 in duplicate wells. After 24 hrs, the culture media was removed, and cells were washed once with DMEM. A 500 μl volume of 50 μM DCFDA in DMEM was added to each well, followed by incubation at 37˚C for 20 min. The cells were washed three times with PBS and DCFDA fluorescence intensities were immediately measured using a fluorimeter (BioTek, Synergy 2) at the excitation wavelength of 485 nm and the emission wavelength of 528 nm at room temperature.

For CSNK1G2 or DUOXA2 overexpression, cells at 70% confluency were transfected with 200 ng plasmids per well in a 24-well plate. After 24 hrs, D4476 was added to the media at a final concentration of 30 μM and DCFDA staining was performed after 24 hrs.

## ROS measurement using Amplex Red

Whole-animal ROS levels were measured using Amplex Red Hydrogen Peroxide/Peroxidase Assay Kit (Invitrogen, A22182) in a 96-well plate following manufacturer's instructions and previous methods with modifications [72,73]. Amplex Red was added to a final concentration of 50 μM. Synchronized L4 animals were incubated at 20˚C for 30 min and measured for fluorescence intensity using a fluorimeter (BioTek, Synergy 2) at the excitation wavelength of 530 nm and the emission wavelength of 590 nm at room temperature. After the measurement, animals in each well were emptied to an agar plate and counted.

To measure dose-dependent effects of D4476 (Sigma, D4476) on ROS levels, HEK293T cells at 70% confluency in 24-well plates coated with PDL were treated with different concentrations of D4476 in duplicate wells. After 24 hrs, the culture media was removed, and cells were washed once with DMEM. A 300 μl volume of 50 μM Amplex Red was added to each well followed by incubation at 37˚C for 20 min and measuring for fluorescence intensity using a fluorimeter (BioTek, Synergy 2) at the excitation wavelength 530 nm and the emission wavelength of 590 nm at room temperature.

## ROS measurement using transgenic HyPer reporter *jrIs1*

Synchronized L4 animals were picked to triplicate wells of a 96-well plate in 100 μl $H_2O$, with 20 animals per well. Fluorescent signals were measured using a fluorimeter (BioTek, Synergy 2) with excitation wavelength of either 490 nm (for oxidized HyPer) or 405 nm (for reduced HyPer) and emission filter of 535 nm. The quantification of the oxidized/reduced HyPer signals was based on the method described by Back et al [60].

## Statistics

*p* values were determined by two-tailed unpaired Student's t-test for comparisons between two samples or Bonferroni multiple comparison with one-way ANOVA for multiple comparison.

## Supporting information

**S1 Fig. CSNK1G protein sequence alignment.** *C.e.*: *C. elegans* CSNK-1; *D.m.*: *Drosophila* Gilgamesh; *M.m.*: mouse CSNK1G2; *H.s.*: human CSNK1G1, CSNK1G2, CSNK1G3. Green bar: frameshift regions caused by *mac494* and *mac495* mutations. Red box: kinase domain. Red arrowhead: *mac397* mutation. Blue box: conserved palmitoylation signal.
(TIFF)

**S2 Fig. Depletion of wildtype *csnk-1* transcripts in L4 *csnk-1(lf)* mutants.** (A) Positions and sequences of PCR primers for detecting all or wildtype-only *csnk-1* transcripts. Partial wildtype and mutant *csnk-1* sequences are aligned to show the specificity of the primers for wildtype-only transcripts. (B) Relative total *csnk-1* transcript levels. (C, D) Relative wildtype-only *csnk-1* transcript levels in wildtype, *csnk-1(mac494lf)* or *csnk-1(mac495lf)* animals. *tba-1* was the loading control. Statistics: two-tailed unpaired Student's *t*-test. *: $p < 0.05$; **: $p < 0.01$; ***: $p < 0.001$.
(TIFF)

**S3 Fig. Phenotypic analyses of *csnk-1(lf)* mutants.** (A) Number of all eggs laid per adult. Results were based three biological replicates, with three animals per replicate. Colors represent different replicates. Statistics: two-tailed unpaired Student's *t*-test. *: $p < 0.05$. ns: not significant. (B) Hatching rate of eggs laid by *csnk-1(lf)* homozygous mutants. All eggs laid by a single adult on regular NGM agar plates were examined. Results were based on three biological replicates, with three animals per replicate. Statistics: two-tailed unpaired Student's *t*-test. ****: $p < 0.0001$. (C, D) Representative molting defects of *csnk-1(lf)* mutants. Arrows indicate attached cuticles. (E) Quantification of young adults (24 hrs after mid-L4 larval stage) with molting defects. Results were based on three biological replicates, with 100 animals analyzed in each replicate. Statistics: two-tailed unpaired Student's *t*-test. **: $p < 0.01$; ***: $p < 0.001$. (F, G) Typical cuticle stripe patterns of *csnk-1(lf)/+* and *csnk-1(lf)* mutants labeled by a DPY-7::sfGFP reporter. (H) Percentage of young adults positively stained by the nuclear dye Hoechst 33258. Results were based on three biological replicates, with 59–141 animals in each replicate. Statistics: two-tailed unpaired Student's *t*-test. ****: $p < 0.0001$. ns: not significant. (I) Colocalization of overexpressed DUOXA2::HA and FLAG::CSNK1G2 in a HeLa cell. (J) Representative fluorescent pictures of wildtype and *csnk-1(lf)* L4 animals stained with DCFDA. Pictures were taken with the same exposure time of 600 ms and fluorescent intensity of each animal was measured using ImageJ. (K) Quantification of DCFDA fluorescent signals of individual L4 animals. Results were based on three biological replicates, with 21–100 animals in each replicate. Statistics: two-tailed unpaired Student's *t*-test. ****: $p < 0.0001$.
(TIFF)

**S4 Fig. Representative morphologies of *csnk-1(lf)*, *bli-3(lf)* and *doxa-1(lf)* single mutants.** All images are of the same scale.
(TIFF)

**S5 Fig. Representative morphologies of double heterozygous mutants between *csnk-1(lf)* and *bli-3(lf)* or *doxa-1(lf)*.** *bli-3(lf) csnk-1(lf)/hT2* (treated as *bli-3(lf) csnk-1(lf)/+ +*) mutants had wildtype-like morphology, while *csnk-1(lf)/hT2; doxa-1(lf)/hT2* (treated as *csnk-1(lf)/+; doxa-1(lf)/+*) mutants exhibited obviously blistered and dumpy phenotype. Arrows point to blisters. All images are of the same scale.
(TIFF)

**S6 Fig. Representative morphologies of double homozygous mutants between *csnk-1(lf)* and *bli-3(lf)* or *doxa-1(lf)*.** *bli-3(lf) csnk-1(lf)* double homozygous mutants were derived from

*bli-3(lf) csnk-1(lf)/hT2* heterozygous mutants. *csnk-1(lf); doxa-1(lf)* double homozygous mutants were derived from *csnk-1(lf)/hT2; doxa-1(lf)/hT2* heterozygous mutants. Arrows point to typical blisters. All images are of the same scale.
(TIFF)

**S7 Fig. CSNK-1::mCherry does not colocalize with GFP and DOXA-1::GFP does not colocalize with mCherry.** (A, B, C) A transgenic L3 larvae co-expressing GFP and CSNK-1:: mCherry in epithelial cells. (D, E, F) A transgenic 3-fold embryo co-expressing DOXA-1::GFP and mCherry in epithelial cells. For unclear reason, DOXA-1::GFP was strongly expressed in embryos but was not visible at larval stages in these transgenic lines. We therefore observed whether DOXA-1::GFP colocalizes with mCherry in embryos.
(TIFF)

**S8 Fig. Sequence alignment of DOXA-1 and its orthologs.** Top three predicted (scansite4. mit.edu) CSNK1 phosphorylation sites (similar to the conserved D/E/(p)S/T(X)$_{1-3}$S/T sequence) in DOXA-1 C-terminal region (start indicated by blue arrow above, predicted by uniprot.org) are enclosed in red boxes with the phospho-acceptor pointed out by red arrowheads. Four potential CSNK1 phosphorylation sites in DUOXA2 C-terminal region (start indicated by blue arrow below, predicted by uniprot.org) are enclosed in green boxes with the phospho-acceptor pointed out by green arrowheads. Two of these sites appear to be conserved in DOXA-1 (purple box and purple arrowheads). *C.e.*: *C. elegans* DOXA-1; *D.m.*: *Drosophila* mol-PF; *M.m.*: mouse DUOXA1; *H.s.*: human DUOXA1 and DUOXA2.
(TIFF)

**S9 Fig. Effects of *csnk-1(lf)* on the expression of *gst-4p::GFP* transgene *dvIs19*.** (A) Synchronized L1 animals of the indicated genotypes were observed after 4 hrs on food. For *csnk-1(lf)* animals, a mixed population of *csnk-1(lf)/hT2; dvIs19* and *csnk-1(lf); dvIs19* were shown, in which the *csnk-1(lf)/hT2* genotype can be identified by pharyngeal GFP signals (arrowheads) expressed from an integrated GFP reporter in *hT2* balancer. Animals without pharyngeal GFPs were identified as *csnk-1(lf)* homozygous (arrows). Pictures were taken with the same exposure time of 100 ms. Total GFP intensity of individual animal was measured using ImageJ and compared with those of *dvIs19* controls. Results were based on over 130 individuals for each genotype. Statistics: two-tailed unpaired Student's *t*-test. ***: $p < 0.001$. (B) Effects of *csnk-1(lf)* on *dvIs19* expression with or without *skn-1(RNAi)*. Animals were treated with feeding RNAi for 72 hrs after hatching. For each experiment, seven animals were aligned and measured for total GFP intensities using ImageJ. The average GFP intensity per animal was adjusted to that of *dvIs19; ctrl RNAi* group. Pictures were taken with the same exposure time of 100 ms. Results were based on three biological replicates. Statistics: two-tailed unpaired Student's *t*-test. *: $p < 0.05$; **: $p < 0.01$.
(TIFF)

**S1 Table. Deleterious genetic variations detected by whole-genome sequencing in the mapped region of the *mac397* isolate.** Ratios of mutated sequences are shown in parentheses. A ratio of 1.00 suggests homozygous, and a ratio less than 1.00 suggests heterozygous. A: adults. ND: not determined.
(TIFF)

**S2 Table. Effects of *csnk-1* mutations or variable feeding RNAis on the Sisi phenotype.** Mammalian orthologs or homologs are indicated in the parentheses.
(TIFF)

**S3 Table. Characterization of *tsp-15(mac499lf)* mutation.** *tsp-15(mac500lf)* and *tsp-15 (mac501lf)* are knockin genocopies of *tsp-15(mac499lf)* generated using the CRISPR/Cas9 method.
(TIFF)

**S4 Table. PCR primers for generating the listed DNA fragments.** Restriction sites are shown in uppercase.
(TIFF)

**S5 Table. PCR primers for generating RNAi plasmids targeting the listed genes.** Restriction sites are shown in uppercase.
(TIFF)

**S6 Table. Genomic target sequences for generating *tsp-15* knockin and *csnk-1* knockout strains using the CRISPR/Cas9 method.** ND: not determined. For *tsp-15* repair template, the letter in red was for introducing the missense mutation and letters in blue were for introducing silent mutations.
(TIFF)

**S1 Text. Strains and constructs.**
(DOCX)

**S1 Raw Data. Raw data for Figures and Tables.**
(XLSX)

## Acknowledgments

We thank Dr Xiaochen Wang for providing the XW18042 *qxIs722* reporter strain. Some strains were provided by the CGC, which is funded by NIH Office of Research Infrastructure Programs (P40 OD010440).

## Author Contributions

**Conceptualization:** Yiman Hu, Zhaofa Xu, Long Ma.

**Data curation:** Yiman Hu, Long Ma.

**Formal analysis:** Yiman Hu, Long Ma.

**Funding acquisition:** Long Ma.

**Investigation:** Yiman Hu, Zhaofa Xu, Long Ma.

**Methodology:** Yiman Hu, Zhaofa Xu, Long Ma.

**Project administration:** Long Ma.

**Resources:** Qian Pan, Long Ma.

**Supervision:** Long Ma.

**Validation:** Yiman Hu, Long Ma.

**Writing – original draft:** Long Ma.

**Writing – review & editing:** Yiman Hu, Zhaofa Xu, Long Ma.

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
