## [Decision Letter · Decision Letter 0]

5 Oct 2022

Dear Dr Ma,

Thank you very much for submitting your Research Article entitled '­­­­Casein kinase 1 gamma regulates oxidative stress response via interacting with the NADPH dual oxidase complex' to PLOS Genetics.

The manuscript was fully evaluated at the editorial level and by independent peer reviewers. The reviewers appreciated the attention to an important problem, but raised some substantial concerns about the current manuscript. Based on the reviews, we will not be able to accept this version of the manuscript, but we would be willing to review a much-revised version. We cannot, of course, promise publication at that time.

If you decide to revise the manuscript for further consideration at PLOS Genetics, please aim to resubmit within the next 60 days, unless it will take extra time to address the concerns of the reviewers, in which case we would appreciate an expected resubmission date by email to plosgenetics@plos.org.

We are sorry that we cannot be more positive about your manuscript at this stage. Please do not hesitate to contact us if you have any concerns or questions.

Yours sincerely,

Danielle A. Garsin

Academic Editor

PLOS Genetics

Gregory P. Copenhaver

Editor-in-Chief

PLOS Genetics

Reviewer's Responses to Questions

**Comments to the Authors:**

Reviewer #1: In this study, Hu, Xu, Pan, and Ma characterize csnk-1 as a novel regulator of oxidative stress in C. elegans. They posit that this regulation occurs via interactions with the bli-3/tsp-15/doxa-1 NADPH dual oxidase genes via nonallelic complementation. The authors employ appropriate methods to study the biological functions of the C. elegans redox system and conservation in mammals. However, the details of the approach, including the number of animals, replicates, and the statistical measure used are not adequately described. Because of this, it is difficult for the reader to assess the power and significance of the reported findings. Mike Murphy and colleagues have expanded upon the inappropriate use of molecules like DCFDA to measure ROS. Given that this is a major component of this manuscript, it is important for the authors to find a more appropriate method to measure ROS, such as the in vivo HyPer reporter which is specific to hydrogen peroxide. Due to missing details regarding the number of replicates and general lack of statistical comparisons, the data in the manuscript are nearly impossible to interpret for both impact and scale of effect. Although the manuscript is appropriate for PLoS Genetics, in its current form, several issues should be addressed prior to publication.

Major Concerns:

• As presented, there is confusion whether the intestine or the hypodermis/epidermis are key in the observed response. This is due to the fact that the data in Table 1 does not support the claims made in the text. It shows that the intestinal expression of csnk-1 rescues survival and not epidermis expression

• In general, experiments need to have appropriate statistical comparisons applied and detailed in the figure legends.

o Table 1/3/4/5 needs to include replicates and p-values

o Data in Figure S2 needs more data than one replicate with a N=3.

• The data as presented in Figure 4b is not appropriately quantified.

• Methodologically the data in Figure 4 seems to be censored by brightness. This quantification needs to be explicitly detailed as it would appear that only the brightest worms are being chosen for analysis as the representative worms. If this is incorrect, the authors should clarify in the methods section how these measurements were performed. In addition, Figure 4a the authors should state what strain or condition the GFP intensity is relative to?

• The authors demonstrate conservation of the interaction between DUOXA2 and CSNK1G2 in mammals, however, this data would be strengthened by expanding the imaging approach in Panel 3A to include more than 1 cell and by performing the reverse IP (IP FLAG and detect HA) to show the biochemical interaction.

Minor Concerns:

• Generally, images should include the relevant exposure times in the figure legend if different across experiments or in the methods section if universally applied.

Reviewer #2: File was attached.

**Have all data underlying the figures and results presented in the manuscript been provided?**

Reviewer #1: Yes

Reviewer #2: Yes

PLOS authors have the option to publish the peer review history of their article (what does this mean?). If published, this will include your full peer review and any attached files.

Reviewer #1: No

Reviewer #2: No

---

## [Decision Letter · Decision Letter 1]

10 Apr 2023

Dear Dr Ma,

We are pleased to inform you that your manuscript entitled "­­­­Casein kinase 1 gamma regulates oxidative stress response via interacting with the NADPH dual oxidase complex" has been editorially accepted for publication in PLOS Genetics. Congratulations!

Yours sincerely,

Danielle A. Garsin

Academic Editor

PLOS Genetics

Gregory P. Copenhaver

Editor-in-Chief

PLOS Genetics

Comments from the reviewers (if applicable):

Reviewer's Responses to Questions

**Comments to the Authors:**

Reviewer #2: The authors honestly responded to the reviewers’ critiques and revised the manuscript well. Taking into account the reviewers’ comments, the authors competently performed additional experiments, remade figures/tables, and mentioned properly in the main text. These point-by-point revisions made this article clear to be understood. I believe this revised manuscript now deserve to be published in PLoS Genetics.

**Have all data underlying the figures and results presented in the manuscript been provided?**

Reviewer #2: Yes

PLOS authors have the option to publish the peer review history of their article (what does this mean?). If published, this will include your full peer review and any attached files.

Reviewer #2: No

**Data Deposition**

http://datadryad.org/submit?journalID=pgenetics&manu=PGENETICS-D-22-01066R1

**Press Queries**

---

## [Editor Report · Acceptance letter]

24 Apr 2023

PGENETICS-D-22-01066R1 

­­­­Casein kinase 1 gamma regulates oxidative stress response via interacting with the NADPH dual oxidase complex 

Dear Dr Ma, 

We are pleased to inform you that your manuscript entitled "­­­­Casein kinase 1 gamma regulates oxidative stress response via interacting with the NADPH dual oxidase complex" has been formally accepted for publication in PLOS Genetics! Your manuscript is now with our production department and you will be notified of the publication date in due course.

With kind regards,

Zsofia Freund

PLOS Genetics

On behalf of:
